# High Concentrations of Circulating 2PY and 4PY—Potential Risk Factor of Cardiovascular Disease in Patients with Chronic Kidney Disease

**DOI:** 10.3390/ijms26094463

**Published:** 2025-05-07

**Authors:** Agnieszka Dettlaff-Pokora, Julian Swierczynski

**Affiliations:** 1Department of Biochemistry, Faculty of Medicine, Medical University of Gdańsk, Dębinki 1, 80-211 Gdańsk, Poland; 2Institute of Nursing and Medical Rescue, State University of Applied Sciences in Koszalin, 75-582 Koszalin, Poland; juls@gumed.edu.pl

**Keywords:** 2PY, 4PY, NAD^+^, nicotinamide, cardiovascular disease, chronic kidney disease

## Abstract

Recently published data indicate that elevated circulating concentrations of N1-methyl-2-pyridone-5-carboxamide (2PY, also described as Met2PY) and N1-methyl-4-pyridone-5-carboxamide (4PY, also described as Met4PY), terminal catabolites of nicotinamide adenine dinucleotide (NAD^+^), are associated with cardiovascular disease (CVD) risk in humans. Previously, we and the others have shown that patients with advanced stages of chronic kidney disease (CKD) exhibit several-fold higher circulating 2PY and 4PY concentrations compared to healthy subjects or patients in the early stages of the disease. It is also well documented that patients with advanced CKD stages exhibit markedly elevated CVD risk, which is the main cause of premature death (in these patients). Therefore, we hypothesize that high concentrations of circulating 2PY and 4PY are important factors that may contribute to cardiovascular events and, ultimately, premature death in CKD patients. However, further, accurately controlled clinical research is needed to provide definitive answers concerning the role of 2PY and 4PY in CVD risk in CKD patients. Moreover, we are dealing with some issues related to the use of NAD^+^ precursors (NAD^+^ boosters) as drugs (also in CKD patients) and/or supplements. Due to the increase in circulating 2PY and 4PY levels during treatment with NAD^+^ boosters, these precursors should be used with caution, especially in patients with increased CVD risk.

## 1. Introduction

In mammals, nicotinamide adenine dinucleotide (NAD^+^) and nicotinamide adenine dinucleotide phosphate (NADP^+^) play two very important functions. One is redox homeostasis—strictly associated with energy production (mainly NAD^+^/NADH) and synthesis of many metabolites and detoxification (mainly NADP^+^/NADPH). The other one is the signaling role [1].

NAD^+^ is reduced to NADH during the catabolism of carbohydrates, lipids, and proteins, primarily in (a) glycolysis, (b) fatty acid β-oxidation, and (c) amino acid oxidation, respectively. NADH formed in these processes serves as a hydride donor for ATP synthesis through oxidative phosphorylation in mitochondria. Moreover, pyruvate formed from glucose during glycolysis or from certain amino acids is transported into mitochondria, where it is oxidized to acetyl-CoA, CO_2_, and NADH. NADH is further oxidized by the mitochondrial respiratory chain, which is associated with ATP production. Acetyl-CoA, formed from pyruvate, fatty acids, and some amino acids, is metabolized in the tricarboxylic acid cycle to CO_2_. The metabolism of acetyl-CoA to CO_2_ is associated with the reduction of NAD^+^ to NADH + H^+^ (by tricarboxylic acid cycle dehydrogenases). NADH formed in the tricarboxylic acid cycle serves as a hydride donor for ATP synthesis via oxidative phosphorylation in mitochondria.

NADPH formed in the phosphogluconate pathway (by glucose 6-phosphate dehydrogenase and 6-phosphogluconate dehydrogenase) and other processes (for instance, in the reaction catalyzed by malic enzyme) serves as a hydride donor in the biosynthesis of many molecules, including fatty acids, cholesterol, steroid hormones, and bile acids. Moreover, NADPH plays an important role in the generation of vascular protective factors and in protection against oxidative stress [2,3].

In addition to its redox role and energy production, briefly presented above, NAD^+^ is a substrate for many other enzymes, including (a) poly(ADP-ribose) polymerase (EC 2.4.2.30); (b) ADP-ribosyltransferase (EC 2.4.2.31); (c) sirtuins (SIRTs)—histone deacetylases; and (d) ADP-ribosyl cyclases. Generally, in these processes, NAD^+^ plays a signaling role.

Poly(ADP-ribose) polymerase (PARP) is a family of enzymes involved in numerous cellular processes, including DNA repair, genomic stability, and programmed cell death [4]. Some PARPs, specifically PARP1, PARP2, PARP5A, and PARP5B, catalyze the transfer of ADP-ribose fragments from NAD+ onto target proteins, forming ADP-ribose chains, as described by the following chemical reaction [1]:(ADP-D-ribosyl)n protein acceptor + NAD^+^ → (ADP-D-ribosyl)n + 1 protein acceptor + nicotinamide + H^+^


In this process, the ADP-D-ribosyl group of NAD^+^ is transferred to the 2′ position of the terminal adenosine moiety, building up a polymer with a chain length of 20–30 units [1].

However, some of the other PARPs catalyze the transfer of only one ADP ribose from NAD^+^ onto target proteins according to the following chemical reaction:protein acceptor + NAD^+^ → (ADP-D-ribosyl) − protein acceptor + nicotinamide + H^+^

Emerging evidence suggests that cellular stress, which is associated with numerous pathologies, triggers accelerated consumption of NAD^+^ via PARP [5,6].

ADP-ribosyltransferase (EC 2.4.2.31), also known as mono-ADP-ribosyltransferase, transfers an ADP-ribose residue from NAD^+^ to a specific protein (specifically onto certain amino acid residues, mainly arginine; however, cysteine and asparagine can also serve as acceptors of the ADP-ribose residue) [7], according to the following reaction:L-arginyl − [protein] + NAD^+^ → N(omega)-ADP-D-ribosyl L-arginyl − [protein] + nicotinamide + H^+^

ADP-ribosylation often leads to specific protein inactivation. Thus, ADP-ribosylation is a molecular mechanism to inhibit protein function [7].

The sirtuin (SIRT) protein family are NAD^+^-dependent histone deacetylases (HDACs) [EC 3.5.1.98]. HDACs remove acetyl groups from N(6)-acetyl-lysine residues on a histone (or other proteins) according to the following reaction:N(6)-acetyl-L-Lysyl [histone or other proteins] + H_2_O → L-Lysyl [histone or other proteins] + acetate

The HDAC family is involved in numerous pathologies [8]. In general, HDACs play an important role in the regulation of transcription and cell proliferation [8]. Moreover, some sirtuins (for instance, SIRT6) can function as NAD^+^-dependent mono-ADP-ribosyltransferases [8].

Overall, in mammals, NAD^+^ is an essential coenzyme in hundreds of oxidoreductases involved in redox reactions related to energy metabolism [9]. Moreover, NAD^+^ is utilized by DNA repair enzymes such as poly(ADP-ribose) polymerase, protein deacylases (sirtuins, SIRTs), ADP-ribosyltransferase, and ADP-ribosyl cyclase [9]. It should be noted that one product of these reactions is nicotinamide (NAM), which can recycle into NAD^+^ or be metabolized to N1-methyl-2-pyridone-5-carboxamide (2PY) and N1-methyl-4-pyridone-5-carboxamide (4PY) (for details, see below).

ADP-ribosyl cyclase CD38 (EC 3.2.2.6), sometimes called an ADP-ribosyl cyclase/cyclic ADP-ribose hydrolase, is a bifunctional enzyme that catalyzes both the synthesis and hydrolysis of cyclic ADP-ribose, a calcium messenger that can mobilize intracellular Ca^2^⁺ stores and activate Ca^2^⁺ influx to regulate a wide range of physiological processes [10].

NAD^+^ deficiency may contribute to numerous pathologies such as metabolic and neurodegenerative diseases, including age-related vascular dysfunction [11]. Aging is the primary condition in which NAD^+^ levels significantly decrease [11]. Moreover, numerous studies have suggested that NAD^+^ levels are related to various chronic metabolic disturbances, including non-alcoholic fatty liver disease, diabetes, kidney diseases, Alzheimer’s disease, and related dementias [12]. It has also been shown that the supplementation of some precursors of NAD^+^ increases intracellular NAD^+^ levels in various tissues and prevents metabolic diseases [12]. Collectively, several precursors of NAD^+^, including nicotinic acid (NA), NAM, and MNAM (N1-methylnicotinamide, see Figure 1), have been used to treat numerous pathologies in both clinical and experimental conditions [12,13]. The results published so far suggest that NAD^+^ precursors can act as potential therapeutic agents in several pathologies, including aging-related body dysfunctions [11,12]. Recently, NAD^+^ homeostasis and the effect of NAD^+^ level and NAD^+^-dependent enzyme activity on chronic low-grade inflammation and improving oxidative metabolism in vascular cells have been reported [14].

In mammals, the catabolism of NAD^+^ and its precursors, including NAM and NA, leads to the formation of numerous catabolites, such as 2PY and 4PY, which can serve as excretory forms of vitamin B_3_ [15]. Under pathological conditions like chronic kidney disease (CKD), circulating 2PY and 4PY are several times higher than in healthy subjects [16,17]. Recently published data suggest that 2PY and 4PY increase cardiovascular risk, independent of traditional CVD risk factors such as high total and LDL cholesterol concentrations, smoking, and diabetes [18]. Thus, one can hypothesize that any factor increasing circulating 2PY and 4PY concentrations in humans, either through elevated synthesis (caused by the increased amount of ingested NAD^+^ precursors) or decreased excretion (for instance, in chronic kidney disease), could be associated with cardiovascular disease (CVD).

In this review, we summarize recent advances regarding NAD^+^ synthesis and degradation, highlighting the role of high concentrations of 2PY and 4PY (the two main NAD^+^ terminal catabolites) in increasing CVD events and premature death in CKD patients. We also hypothesize that NAD^+^ precursors, such as NA, which has been used (and is still used) as an effective drug in CVD risk prevention, and NAM, applied in the treatment of hyperphosphatemia in CKD patients, may increase 2PY and 4PY levels and consequently increase CVD risk. Therefore, we suggest that NAD^+^ precursors, as potential therapeutic agents and/or dietary supplements, should be used with caution, especially in patients with increased CVD risk.

## 2. NAD^+^ Precursors Ingested by Humans

NAM is an essential component of NAD^+^ and NADP^+^. In mammals, approximately 10% of intracellular NAD^+^ is converted to NADP^+^ in the reaction NAD^+^ + ATP → NADP^+^ + ADP, catalyzed by NAD kinase (EC 2.7.1.23). In turn, NADP^+^ can be dephosphorylated to NAD^+^ (according to the reaction NADP^+^ + H_2_O → NAD^+^ + Pi) by NADP^+^ phosphatases [19,20].

NAM, the predominant form of vitamin B_3_, is a water-soluble amide derivative of NA. Vitamin B_3_ is a common term for NA, NAM, and their derivatives that exhibit the biological activity of NAM. All these compounds are often called niacin, although niacin may also refer specifically to NA. Sometimes, vitamin B_3_ is also defined as the dietary NAD^+^ precursor, other than tryptophan. Vitamin B_3_ is generally highly bioavailable from foods of both animal and plant origin; however, bioavailability from some plants is relatively lower. It is mainly found in meat (beef and poultry), liver, fish, legumes, nuts, grain products, vegetables, and some cereals [21]. Vitamin B_3_ deficiency is generally uncommon in developed countries, except for specific groups in the population (e.g., vegans, patients affected by eating disorders, Crohn’s disease, gastrointestinal infections, excessive alcohol use, or cancer). Deficiency, if it occurs, causes pellagra, an endemic disease associated with dementia, diarrhea, and dermatitis [22]. The storage of this vitamin in the human body is minimal, and excess is rapidly excreted, primarily in urine. Therefore, vitamin B_3_ needs to be constantly supplied by the diet to maintain homeostatic levels required for normal metabolism. Importantly, plasma vitamin B_3_ concentrations are markedly higher in children compared to adults [23]; however, the mechanisms responsible for this phenomenon are not yet fully understood.

## 3. NAD^+^ Synthesis Pathways in Mammals

The data presented above clearly indicate that NAD^+^ plays a vital role in mammals. Therefore, the intracellular levels of NAD^+^ must be tightly regulated by its biosynthesis and consumption processes. Disruption of this homeostasis could lead to significant changes in intracellular NAD^+^ levels and, consequently, to disturbances in many biochemical and physiological processes. For instance, some results suggest that dysregulation of NAD^+^ metabolism might contribute to obesity, metabolic syndrome, diabetes, and diabetic complications [24]. A significant decrease in intracellular NAD^+^ levels has been found in aging, which is closely linked to a reduction in mitochondrial function [25]. Therefore, it is tempting to speculate that the decrease in intracellular NAD^+^ concentration is associated, at least in part, with the markedly lower levels of plasma vitamin B_3_ in adults compared to children [23].

In mammals, NAD^+^ is synthesized through two processes: (a) from NAM, NA, NR (nicotinamide ribose), and nicotinamide mononucleotide (NMN) (via a process called the salvage pathway; see Figure 2), and (b) from dietary tryptophan (via a process called the de novo pathway; see Figure 3).

As already mentioned, NAM is primarily found in meat, milk, eggs, fortified cereals, and some supplements [26]. It is formed in cells by enzymes that consume NAD^+^. NA is primarily found in meat, grains, fortified cereals, and supplements [26]. NR and tryptophan (as a protein component) are found in meat, eggs, fish, and cheese [26]. Finally, NMN can be obtained through supplements [26]. It should be noted that among the dietary precursors of NAD^+^, only NA is associated with a transient vasodilatory reaction (flushing induction), caused by COX-2-dependent prostanoid formation [27].

In mammalian cells, the main precursor of NAD^+^ is nicotinamide (NAM) [20], which is converted to NMN by nicotinamide phosphoribosyltransferase (NAMPT), the rate-limiting enzyme in the NAD^+^ biosynthesis process, according to the reaction NAM + PRPP → NMN + PPi [28]. In turn, NMN is converted to NAD^+^ by nicotinamide mononucleotide adenyltransferase (NMNAT), which catalyzes the reaction NMN + ATP → NAD^+^ + PPi [29]. It should be noted that stimulation of NMNAT activity leads to increased intracellular NAD^+^ levels [30]. NMN can also be formed from NR directly in the reaction NR + ATP → NMN + ADP, catalyzed by riboside kinase [31], or indirectly into NAM by purine nucleoside phosphorylase [32]. Finally, NAM is converted to NAD^+^ as described above. Moreover, NAM produced in the liver is used in this organ or can be released into the serum and then taken up by other organs as a substrate for NAD^+^ biosynthesis in the salvage pathway [33].

NA is converted to nicotinic acid mononucleotide (NAMN) in the reaction NA + PRPP → NAMN + PPi, catalyzed by nicotinic acid phosphoribosyltransferase (NAPRT) [29]. NAMN, in turn, is converted to nicotinic acid adenine dinucleotide (NAAD) in the reaction 2NAMN + ATP → NAAD + PPi. In the final step, NAAD is converted to NAD^+^ in a reaction catalyzed by glutamine-dependent NAD^+^ synthetase. This enzyme catalyzes the following reaction: Deamido-NAD^+^ (NAAD) + L-glutamine + ATP + H_2_O → NAD_+_ + L-glutamate + AMP + PPi. Figure 2 presents NAD^+^ (and NADP^+^) synthesis through the salvage pathway. It should be noted that in the literature, the conversion of NA to NAMN and further to NAAD is referred to as the Preiss–Handler pathway [13].

It should be added that NAD^+^ content in endothelium can be modulated also by the extracellular conversion of NMN and NR by CD73 [34].

In the liver, which is the only organ that possesses all the enzymes required for de novo NAD^+^ synthesis, tryptophan is metabolized to quinolinic acid (QA). In the first step of de novo NAD^+^ synthesis, tryptophan is converted to N-formylkynurenine by tryptophan-2,3 dioxygenase (primarily in the liver) or by indoleamine 2,3 dioxygenase in extrahepatic tissues, including endothelial cells [35]. Next, N-formylkynurenine is converted through a series of steps (kynurenine → 3-hydroxykynurenine → 3-hydroxy-anthranilic acid → aminocarboxymuconic semialdehyde → quinolinic acid) to NAMN [36]. Finally, NAMN is converted to NAD^+^ via NAAD (Figure 3).

An important regulator of the conversion of tryptophan to NAD^+^ is α-amino-β-carboxy-muconate semialdehyde decarboxylase (ACSMD), a key checkpoint in de novo NAD^+^ synthesis. This enzyme converts the unstable aminocarboxymuconic semialdehyde (ACMS) to aminomuconic semialdehyde (AMS). AMS can then be enzymatically converted to CO_2_ (via glutaryl-CoA → acetyl-CoA → tricarboxylic cycle) or non-enzymatically to picolinic acid, which is subsequently converted to NAD^+^ (Figure 3). Low ACSMD activity allows ACMS to convert non-enzymatically to quinolinic acid, eventually leading to NAD^+^ synthesis. In contrast, elevated ACSMD activity prevents the conversion of ACMS to NAD+ by shifting the pathway towards AMS formation.

Importantly, pharmacological inhibition of ACSMD has been shown to increase de novo NAD+ synthesis, enhance mitochondrial function (evidenced by increased expression of genes encoding citrate synthase and enzymes involved in oxidative phosphorylation), and improve health [37]. Thus, ACSMD could be a potential drug target to influence intracellular NAD^+^ levels.

It should be noted that the synthesis of NAD^+^ from NAM requires only two steps, while from NA, it requires three steps (Figure 2). In contrast, the de novo synthesis of NAD+ from tryptophan follows an eight-step pathway (Figure 3). Interestingly, deficiency of quinolinate phosphoribosyltransferase, a key enzyme in converting tryptophan to NAD^+^, does not significantly affect NAD^+^ levels in the liver or other mouse organs [38].

Thus, it can be concluded that in mammals, NAD^+^ synthesis predominantly occurs through the salvage pathway. However, some evidence suggests that the de novo NAD^+^ synthesis pathway becomes more active, particularly during aging and inflammation [39]. Therefore, it can be assumed that NAD^+^ synthesis in mammals can switch between the salvage and de novo pathways depending on pathophysiological conditions.

## 4. NAD^+^ and NAM Catabolism and Excretion of Formed Catabolites

Under physiological conditions, NAM that is not recycled into NAD^+^ is primarily metabolized into N1-methyl-2-pyridone-5-carboxamide (2PY) and N1-methyl-4-pyridone-5-carboxamide (4PY) (Figure 4). This process occurs in the cytosol and starts with the methylation of NAM to form N1-methylnicotinamide (MNAM; see Figure 4). The enzyme responsible for this reaction is nicotinamide N-methyltransferase (NNMT), which uses S-adenosyl-L-methionine (SAM) as a methyl donor [40]. The reaction catalyzed by this enzyme is as follows:NAM + SAM → MNAM + SAH
where SAM—S-adenosylmethionine and SAH—S-adenosylhomocysteine.

It should be noted that NNMT activity can affect intracellular NAD^+^ level and, consequently sirtuin activity, which exerts an important role in the prevention of vascular aging [41].

In turn, N1-methylnicotinamide (MNAM) is further oxidized by aldehyde oxidase into two related compounds: N1-methyl-2-pyridone-5-carboxamide (2PY) and N1-methyl-4-pyridone-5-carboxamide (4PY). Both MNAM and its metabolites, 2PY and 4PY, formed during the catabolism of NAM, can be excreted in urine [40].

The scheme of the conversion of NAM (formed from NAD^+^ and NAD^+^ precursors) to 2PY, 4PY, and nicotinamide N-oxide is presented in Figure 4.

Moreover, in the human liver endoplasmic reticulum, NAM is oxidized to nicotinamide N-oxide (3-pyridinecarboxamide 1-oxide), mainly by the enzyme CYP2E1 [42]. Nicotinamide N-oxide is also eliminated in the urine [40] (Figure 4). It is important to note that CYP2E1 is also involved in ethanol metabolism, precisely in the conversion of ethanol to acetaldehyde [43]. This suggests that excessive ethanol consumption might interfere with NAM (and NAD^+^) metabolism, particularly the conversion of NAM to nicotinamide N-oxide. Under physiological conditions, conversion of NAM to 2PY and 4PY is quantitatively predominant in NAM (and its precursors, including NAD^+^) catabolism and elimination. However, when very high levels of NA are used for treating hyperlipidemia, NAM is also metabolized to nicotinamide N-oxide [44].

## 5. Potential Pathophysiological Function of NAD^+^ Metabolites in CKD Patients

Recently, it has been shown that in humans, two terminal metabolites of NAD^+^, nicotinamide and nicotinic acid—namely, 2PY and 4PY—are clinically associated with cardiovascular disease (CVD), independent of traditional risk factors [18]. Moreover, genome-wide association studies and other functional studies indicate that a variant within the ACSMD gene (the gene encoding α-amino-β-carboxy-muconate semialdehyde decarboxylase, which controls the rate of conversion of tryptophan to NAD^+^; see Figure 3) is significantly associated with blood concentrations of 2PY and 4PY and with sVCAM-1 levels [18]. Furthermore, 4PY (but not 2PY) enhances endothelial cell activation and VCAM-1 gene expression (measured both at the mRNA and protein levels) [18]. These results suggest that elevated circulating levels of 2PY and, especially, 4PY—terminal catabolites of NAD^+^, NAM, NA, and other precursors of NAD^+^—could be linked to the pathogenesis of CVD, possibly via an inflammatory pathway that increases endothelial cell VCAM-1 gene expression.

Thus, it is likely that any factor increasing circulating 2PY and 4PY concentrations (either through elevated synthesis or decreased excretion) may be associated with cardiovascular disease (CVD). Chronic kidney disease (CKD) is a pathology in which circulating 2PY and 4PY are several times higher, both in adults and children, and this condition typically persists for many years [16,45,46,47,48]. A significant positive correlation was found between plasma 2PY and 4PY levels and creatinine concentration (marker of renal function). Moreover, (a) a single hemodialysis treatment was associated with a significant, though temporary, reduction in plasma 2PY and 4PY, and (b) successful kidney transplantation was associated with the return of circulating 2PY and 4PY concentrations to values similar to those present in healthy subjects. The elevated concentrations of circulating 2PY and 4PY in CKD patients are possibly due to impaired elimination of metabolites by the kidneys [16,49]. Additionally, using an experimental model of CKD, we found that 2PY and 4PY accumulate not only in the blood but also in various organs of uremic rats [16]. Thus, it is very likely that similar changes occur in CKD patients. We have also shown that 2PY inhibits PARP activity in vitro [16]. Based on these data, more than 20 years ago, we proposed that 2PY could be a uremic toxin [16]. However, thus far, no clear data have been published indicating that terminal NAD^+^ catabolites are harmful to the human body. To the best of our knowledge, only one study has shown that patients with the highest circulating 2PY concentrations have a higher risk of initiating dialysis [50]. This effect has not been confirmed after adjustments for age or CKD stages [50]. Therefore, further studies involving large groups of CKD patients are needed to resolve this issue.

If 2PY and 4PY are indeed responsible for the increased CVD risk, as suggested by the authors of the aforementioned study [18], one would expect that such a condition (i.e., elevated levels of 2PY and 4PY lasting for several years) would be very harmful to CKD patients. Recent findings indicate that elevated circulating concentrations of 2PY and 4PY are strongly associated with an increased risk of CVD [18]. Therefore, one could conclude that these results may provide an explanation for the association between CKD and CVD in CKD patients. It is well documented that patients with CKD exhibit elevated cardiovascular risk, including coronary artery disease, heart failure, arrhythmias, and sudden cardiac death [51]. Moreover, patients with advanced stages of CKD exhibit a markedly higher CVD risk, which is the leading cause of premature death in these individuals [51]. It should be noted that patients with advanced CKD stages have significantly elevated circulating 2PY and 4PY concentrations compared to those in the early stages of the disease [16,50]. Furthermore, CKD is associated with a chronic inflammatory state, which contributes to vascular and myocardial remodeling processes [51]. Thus, it is tempting to speculate that elevated levels of circulating 2PY and 4PY might contribute to cardiovascular events in CKD patients. Moreover, the findings suggest that elevated concentrations of 2PY and 4PY could contribute to CVD via increased VCAM1 gene expression and leucocyte adherence to the vascular wall and, finally, vascular inflammation (a critical component of CVD) [18] supports the hypothesis that 2PY could indeed be a uremic toxin [16]. Figure 5 presents a scheme illustrating the proposed mechanistic hypothesis of high concentrations of 2PY and 4PY in CKD on CVD.

Collectively, the results showing high circulating 2PY and 4PY concentrations in CKD patients, along with the identification that these catabolites increase cardiovascular risk [18], help explain why patients with advanced stages of CKD exhibit markedly elevated CVD risk, which is the leading cause of death in these patients. However, whether high circulating 2PY and 4PY truly contribute to CVD and excess mortality (or other adverse effects) in advanced stages of CKD remains to be determined in future studies.

The general view on the role of elevated 2PY and 4PY in CKD patients presented above seems to contrast with recent findings published by Yoshimura et al. [52]. They suggested that terminal NAD^+^ catabolites, including 2PY, are not uremic toxins but may be potential therapeutic compounds that inhibit fibrosis and inflammation in a fibrotic kidney mouse model [53]. Their conclusion is primarily based on the following observations:

(a) 2PY, 4PY, and nicotinamide N-oxide inhibit TGFβ1-induced fibrosis and inflammatory gene expression in kidney fibroblasts; (b) 2PY suppresses the expression of genes encoding collagen (types I and III), αSMA, and IL-6 in kidney fibroblasts; (c) 2PY inhibits TGFβ1-induced gene expression encoding collagen (type I) and IL-6 in renal tubular epithelial cells; (d) no toxic effect of 2PY was observed in UUO (unilateral ureteral obstruction) mice; and (e) the chemical structure of 2PY and 4PY shares similarities with pirfenidone, a drug that inhibits collagen synthesis and fibroblast proliferation, used to treat idiopathic pulmonary fibrosis [54] and diabetic kidney disease [55].

Moreover, it has been proposed that 2PY inhibits the NLRP3 pathway and macrophage infiltration in inflammatory diseases, similar to pirfenidone [53]. NLRP3 is a multiprotein complex that plays a crucial role in regulating the innate immune system and inflammatory signaling in response to various stimuli [56]. One possible explanation for the discrepancy discussed above could be the experimental model used by Yoshimura et al. [53]. Most of the results presented by Yoshimura et al. were obtained using cell cultures (kidney fibroblasts, renal tubular epithelial cells). Furthermore, the animals in Yoshimura et al.’s experiments were treated with 2PY for only 7 days. It seems that a 7-day exposure to 2PY is too short a period to induce conditions that would enable the development of cardiovascular disease (CVD). Therefore, it may be premature to conclude that 2PY is not a uremic toxin based on such a limited timeframe.

Finally, the question arises: can we prevent high concentrations of 2PY and 4PY to ultimately limit the development of CVD in CKD patients? From a clinical point of view, this is a very important question. Our previously reported data indicate that the most effective way to decrease circulating 2PY and 4PY concentrations in CKD patients is successful kidney transplantation, which reduces these concentrations to control levels [16]. Moreover, it has been shown that successful kidney transplantation is the most effective intervention for reducing CVD risk in kidney transplant recipients [57]. However, it should be emphasized that although CVD risk is significantly reduced, it remains the leading cause of morbidity and mortality in kidney transplant recipients [58]. Overall, these findings support, at least in part, the view that elevated circulating levels of 2PY and 4PY in CKD patients may be only partially, if at all, responsible for the development of CVD in this population.

A temporary decrease in circulating 2PY and 4PY concentration in CKD was also observed in hemodialysis patients (2PY and 4PY concentrations decreased just after and increased significantly 48 h after the end of treatment) [16]. Therefore, one could expect some positive effect of hemodialysis on CVD risk. However, it has been shown that the incidence of CVD in patients with CKD may increase several-fold in dialysis patients [59,60,61]. This is related to many different factors, including dialysis-specific factors such as dialysis catheters, membrane exposure, endotoxemia, and more rapid loss of residual kidney function. All these factors may contribute to inflammation and oxidative stress, which may ultimately increase CVD risk in hemodialyzed patients irrespective of circulating 2PY and 4PY concentrations. For this reason, a temporary decrease in circulating 2PY and 4PY cannot decrease CVD risk in CKD patients.

Since NNMT plays an important role in 2PY and 4PY synthesis (see Figure 4), one can conclude that inhibition of this enzyme may lead to a decrease in the circulating concentration of 2PY and 4PY. Thus, it is tempting to speculate that NNMT inhibitors can be used in clinical practice, for instance by CKD patients. Notably, several NNMT inhibitors are already available and were suggested as a promising strategy for some pathological conditions such as cancer, diabetes, obesity, and neurodegenerative diseases [62,63,64].

## 6. Beneficial and Possible Adverse Side Effects of NAD^+^ Precursors (NAD^+^ Boosters) Used as Therapeutic Agents or Supplements

Preclinical and clinical observations suggest that increasing intracellular NAD^+^ concentrations is linked to improvements in the physiological function of the human body. NAD^+^ cannot be administered orally because it does not cross cell membranes. Therefore, elevated intracellular concentrations can be achieved by supplementing the diet with its precursors, mainly NAM, NA, NR, NMN, and tryptophan. Numerous preclinical studies indicate the potential effectiveness of NAD^+^ precursors, including NAM, NMN, NR, and NA supplementation, in preventing conditions such as (a) dementia, (b) stroke, (c) traumatic brain injury, and (d) diabetes [65]. Moreover, it has been reported that NMN can improve the imbalance of the NAD^+^/NADH ratio and recruit GSH to enhance GPX4-mediated ferroptosis defense in UV-induced skin injury [66].

Several human clinical trials also have investigated the potential effectiveness of NAD^+^ precursors. In general, oral administration of NR to healthy volunteers has been shown to increase (a) NAD^+^ and NAAD levels in peripheral blood mononuclear cells (PBMCs), (b) NADH and NADPH levels in red blood cells (RBCs), and (c) NAD^+^ concentration in whole blood. Moreover, some positive physiological effects were observed after NR administration, with no serious adverse side effects reported [13]. In obese men with metabolic disorders, oral administration of NR led to increased urinary concentrations of NR, NAM, and MNAM. However, no improvement in insulin sensitivity, glucose production, glucose disposal, or glucose oxidation was observed [13]. Notably, no serious adverse side effects of NR administration were found in this group either. Additionally, oral administration of NR combined with pterostilbene in healthy volunteers resulted in increased NAD^+^ levels in whole blood. However, this group also exhibited an increase in serum total and LDL cholesterol concentrations [13].

The results presented so far suggest that NAD^+^ precursors can serve as potential therapeutic agents for various pathologies, including aging-related dysfunctions [11]. However, due to potential adverse side effects—such as an increased risk of CVD [18] and the inhibition of PARPs and sirtuins—both NA and NAM should be used with caution [67].

Recently published results indicate that NMN supplementation increases skeletal muscle insulin sensitivity in prediabetic and obese female adults [68] and has a positive impact on physiological endurance and overall health in healthy adults [69]. Moreover, it has been shown that oral administration of NMN, up to 900 mg per day for 60 days, is safe and well tolerated [69]. The safety and anti-aging effects of NMN in human clinical trials have also been recently reported. These findings suggest that NMN supplementation increases NAD^+^ concentrations and may help mitigate aging-related disorders (including oxidative stress, DNA damage, neurodegeneration, and inflammatory responses) [70]. Additionally, NMN administration (2 MIB-626 tablets each containing 500 mg of NMN twice daily for 28 days) in overweight or obese patients (middle-aged and older adults) led to an increase in circulating NAD^+^ catabolites, including 2PY [52]. The increased circulating concentrations of NAD^+^ and 2PY were associated with reduced (a) circulating total and LDL cholesterol and non-HDL cholesterol concentrations, (b) body weight, and (c) diastolic blood pressure [52]. Although these results are interesting, the interpretation of these findings is limited by its small sample size (30 patients) and short intervention duration [52].

It should be noted that the administration of NAD^+^ precursors to patients has been shown to increase circulating NAD^+^ catabolite ls including 2PY [18,52,68,71,72,73]. This raises some concerns regarding the safety of using NAD^+^ boosters.

NAM is readily absorbed from the human gastrointestinal tract, with peak concentrations achieved approximately one hour after oral ingestion [74]. It has been used for many years to treat various disorders, including type 1 diabetes [74]. Based on numerous clinical studies, the general conclusion is that NAM is safe; however, higher doses of this drug should be considered as potentially detrimental [74]. Some data suggest that NAM and its metabolite MNAM, when administered in high doses before alloxan exposure, can protect mice from alloxan-induced diabetes [75]. In vitro studies using isolated peritoneal mouse macrophages have demonstrated that NAM possesses anti-inflammatory properties [76]. These properties are attributed to NAM’s ability to reduce the synthesis of several inflammatory mediators, including TNFα, IL-6, IL-10, IL-12p40, NO, and PGE2 [76].

NAM inhibits active intestinal phosphate transport and has been used to treat hyperphosphatemia in patients with CKD [17]. Its therapeutic effect is comparable to that of sevelamer (SEV), an orally administered polymer/resin that binds dietary phosphate in the gastrointestinal tract. SEV is commonly used by CKD patients on hemodialysis or peritoneal dialysis to manage hyperphosphatemia [77]. Both drugs are equally effective in reducing circulating phosphate concentrations. However, patients treated with NAM exhibit very high circulating levels of 2PY, which may contribute to side effects such as nausea, diarrhea, and thrombocytopenia—these adverse effects occur more frequently in the NAM group compared to the SEV group [17]. These findings, particularly the occurrence of thrombocytopenia, suggest that high concentrations of 2PY could be detrimental. Interestingly, thrombocytopenia has also been observed in clinical studies on niacin treatment [78]. Notably, within one week of discontinuing niacin, platelet counts significantly increase, returning to control levels approximately five weeks after drug cessation [78].

Some data suggest that MNAM (N1-methylnicotinamide) exerts antithrombotic activity [79] and prevents endothelial dysfunction in hypertriglyceridemic and diabetic rats [80]. It has been proposed that the anti-diabetic effects of MNAM may be linked to its vasoprotective activity [81]. Unlike NAM, the anti-inflammatory effects of MNAM are primarily associated with its action on the vascular endothelium [76]. Furthermore, Brzozowski et al. demonstrated the therapeutic potential of MNAM in protecting against acute gastric lesions induced by stress [82].

Very recently the list of clinical trials on NAM in dermatological disorders has been published [83]. These studies indicate the potential application of NAM in preventing and treating numerous skin pathologies. Based on the anti-inflammatory actions of NAM and MNAM, these compounds have been explored as treatments for certain skin diseases. NAM was used to treat psoriasis [84] and rosacea (chronic facial dermatosis) [85]. Recently published data indicate that several studies are underway to explore the potential application of nicotinamide in preventing and treating various skin cancers and other skin pathologies [83].

Moreover, alterations in circulating MNAM concentrations and urinary excretion have been reported in numerous pathological conditions. For instance, serum and urinary MNAM levels were significantly increased in patients with liver cirrhosis [86]. Additionally, urinary excretion of MNAN and 2PY was also significantly elevated in these patients [86]. Urinary excretion of MNAM in patients with affective disorders was significantly higher than in healthy controls [87]. In patients with Parkinson’s disease, urinary excretion of MNAM was also significantly higher than in healthy controls [88]. Furthermore, the urinary excretion of MNAM and 2PY was significantly increased in children with burns after injury [89].

Treatment with NR in patients with ataxia telangiectasia leads to improvement in patients’ states [73].

Niacin has been in clinical use for more than 50 years [90] as an effective drug for CVD risk prevention [91,92]. Notably, niacin was the first drug shown to reduce cardiovascular events and mortality in patients with myocardial infarction [93]. Niacin may exert these favorable effects by reducing total cholesterol, LDL cholesterol, TAG, and lipoprotein (a), while simultaneously increasing HDL cholesterol levels [94,95]. However, some clinical data suggest that niacin provides health benefits through mechanisms independent of changes in circulating lipids [95,96]. These lipid-independent benefits include anti-inflammatory and antioxidant effects [97]. Regarding its anti-inflammatory properties, niacin downregulates several mediators of vascular inflammation, such as VCAM-1, MCP-1, and P-selectin [98]. Additionally, niacin increases the expression and release of adiponectin, a protein that exerts protective effects against vascular inflammation and endothelial dysfunction [99,100]. Niacin also reduces endothelial oxidative stress by increasing intracellular NADPH and GSH levels and by inhibiting reactive oxygen species production in endothelial cells [101]. Furthermore, the addition of niacin to statin therapy has shown favorable combined effects on various surrogate endpoints [102,103,104,105,106,107].

However, some clinical trials have indicated that while niacin therapy significantly increases serum HDL cholesterol concentrations, this is not associated with a reduction in recurrent cardiovascular events such as myocardial infarction, stroke, or revascularization [108]. Moreover, niacin therapy has been shown to be associated with an increased risk of new-onset or worsening diabetes, as well as skin, gastrointestinal, and musculoskeletal adverse effects [108].

Niacin (which is often fortified in food staples) contributes to NAD^+^ biosynthesis and increases blood concentration of 2PY and 4PY when it is consumed in excess [109,110,111].

In a systematic review and meta-analysis of seven randomized controlled trials involving 441 participants (228 in the niacin group and 213 in the control group), Sahebkar showed that niacin supplementation improves endothelial function [112]. However, a systematic review and meta-analysis of 23 randomized controlled trials (published between 1968 and 2015), including approximately 40,000 participants, showed that supplementation with NA (median dose of 2 g per day) had no significant effect on total mortality, cardiovascular mortality, non-cardiovascular mortality, acute myocardial infarction, and stroke [113]. In a systematic review and meta-analysis of five randomized controlled trials, 230 patients were involved. He et al. reported that niacin (0.375–1 g per day) reduced hyperphosphatemia in CKD patients similarly to NAM [114].

It has also been reported that higher niacin intake could have a protective effect on the development of cognitive decline (Alzheimer’s disease and other forms of dementia) [65,115].

Numerous meta-analyses of clinical trials on niacin supplementation have reported adverse effects. For instance, participants with or at risk for CVD who received niacin (0.5–4 g per day) had to discontinue treatment due to side effects such as flushing, pruritus, rash, gastrointestinal problems, and new-onset diabetes [113]. Similar adverse side effects, particularly flushing, were also observed in patients with renal disease treated with niacin (0.375–1.5 g per day).

Current knowledge and future directions regarding dietary supplementation with NAD^+^ precursors in humans have been recently summarized and discussed [12]. The general conclusion drawn from the results presented in this review is that NAD^+^ precursors are safe, well-tolerated, and increase intracellular NAD^+^ concentrations. However, the authors of this review emphasized that numerous human trials to date have not demonstrated clinically significant health benefits following treatment with NAD+ boosters. It is likely that the relatively small sample sizes studied so far limit data interpretation.

The data discussed above indicate that precursors of 2PY and 4PY, including niacin, NAM, and MNAM, have been used to treat numerous pathologies in both clinical and experimental conditions. Moreover, the aforementioned compounds can increase 2PY and 4PY levels [18,68,71,72]. Therefore, such therapy could be associated with an increased risk of CVD. Collectively, NAD^+^ precursors (NAD^+^ boosters) can be used as drugs to prevent and/or treat certain diseases; however, some data presented in this review suggest that higher doses of these compounds administered to individuals who may already have an increased risk of CVD could be considered detrimental after conversion to 2PY and 4PY, which are associated with cardiovascular disease (CVD). Thus, one can conclude that NAD^+^ boosters may have a primarily beneficial effect on human health. However, when NAD^+^ boosters are used for a long time in amounts that lead to several-fold increase in circulating 2PY and 4PY, they could be detrimental to human health, especially to the cardiovascular system.

## 7. Conclusions

Recent observations indicating that N1-methyl-2-pyridone-5-carboxamide (2PY) and N1-methyl-4-pyridone-5-carboxamide (4PY) are associated with cardiovascular disease risk in humans suggest that high concentrations of circulating 2PY and 4PY are important factors that may contribute to cardiovascular events and, ultimately, premature death in CKD patients. Therefore, 2PY and 4PY can be regarded as genuine uremic toxins. However, the results presented and discussed in this review only pave the way for further research on the role of 2PY and 4PY in CVD risk in CKD patients.

Moreover, in this review, we discussed the efficacy and benefits of NAD^+^ precursors (NAD^+^ boosters) such as nicotinic acid, nicotinamide, nicotinamide riboside, and nicotinamide mononucleotide, which can play protective roles in certain diseases and age-related pathologies. We also emphasized that NAD^+^ precursors may have primarily beneficial effects; however, detrimental effects could also be observed. The beneficial effect could be associated with increased intracellular NAD^+^ levels, a substrate for critical cellular processes including energy production, DNA repair, maintenance of genomic integrity, gene expression, epigenetic regulation, calcium signaling, innate immune response, and inflammation. The potential detrimental effect could be related to the overproduction and accumulation of 2PY and 4PY while excessively intaking NAD^+^ precursors. These two compounds, in light of recently published results, are associated with cardiovascular disease (CVD) risk in humans, and their circulating concentrations increase after treatment with NAD^+^ precursors. Therefore, NAD^+^ precursors (NAD^+^ boosters) should be used with caution, especially in patients with an increased risk of CVD.

## Figures and Tables

**Figure 1 ijms-26-04463-f001:**
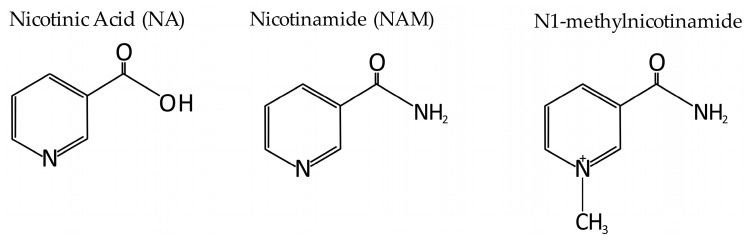
Chemical structure of nicotinic acids (NA), nicotinamide (NAM), and N1-methylnicotinamide (MNAM).

**Figure 2 ijms-26-04463-f002:**
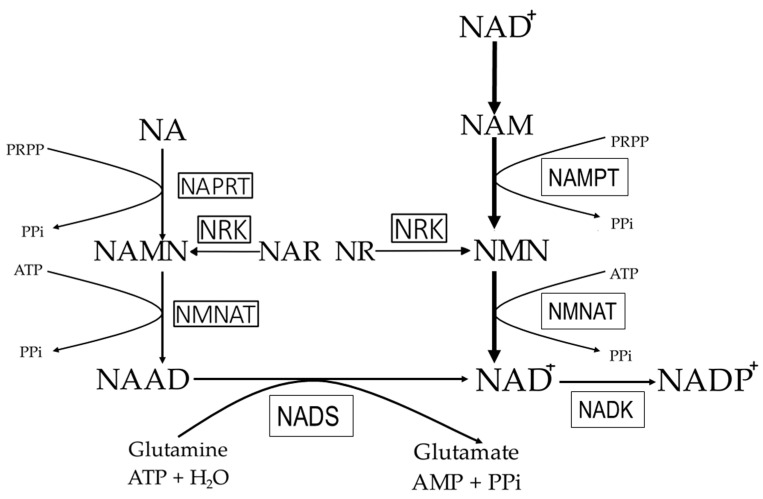
NAD^+^ synthesis through the salvage pathway. NA—nicotinic acid; NAAD—nicotinic acid adenine dinucleotide; NAD^+^—nicotinamide adenine dinucleotide; NADP^+^—nicotinamide adenine dinucleotide phosphate; NAM—nicotinamide; NAMPT—nicotinamide phosphoribosyltransferase; NADK—NAD^+^ kinase; NADS—glutamine-dependent NAD^+^ synthetase; NAPRT—nicotinate phosphoribosyltransferase; NMN—nicotinamide mononucleotide; NMNAT—nicotinamide mononucleotide adenylyltransferase; NRK—nicotinamide riboside kinase.

**Figure 3 ijms-26-04463-f003:**
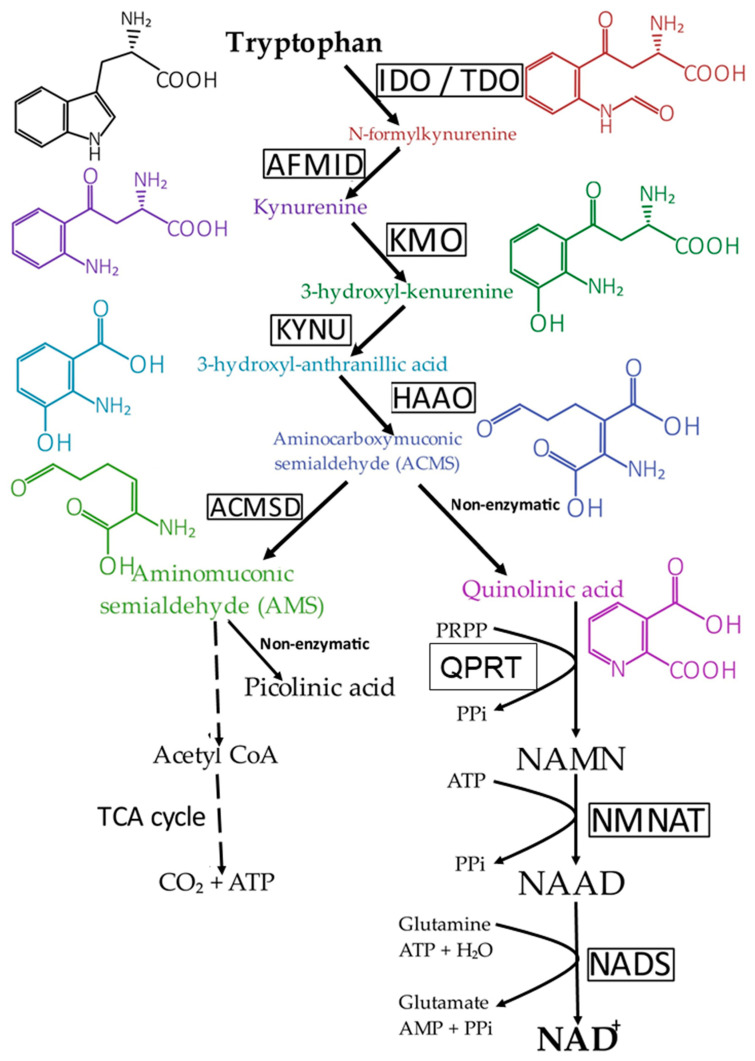
NAD^+^ synthesis through the de novo pathway. ACMSD—aminocarboxymuconic semialhedyde decarboxylase; AMID—N-formylkynurenine formamidase; AMS—aminomuconic semialdehyde; HAAO—3-hydroxyanthranilate 3,4-di-oxygenase 2; IDO—indoleamine 2,3-dioxygenase; KMO—kynureine 3-monooxygenase; KYNU—kynureninase; NAAD—nicotinic acid adenine dinucleotide; NAD^+^—nicotinamide adenine dinucleotide; NADS—glutamine-dependent NAD^+^ synthetase; NAMN—nicotinic acid mononucleotide; NMNAT—nicotinamide mononucleotide adenylyltransferase; PPi—pyrophosphate; PRPP—phosphoribosyl pyrophosphate; QPRT—quinolate phosphoribosyl transferase; TCA—tricarboxylic acid cycle; TDO—tryptophan 2,3-dioxygenase.

**Figure 4 ijms-26-04463-f004:**
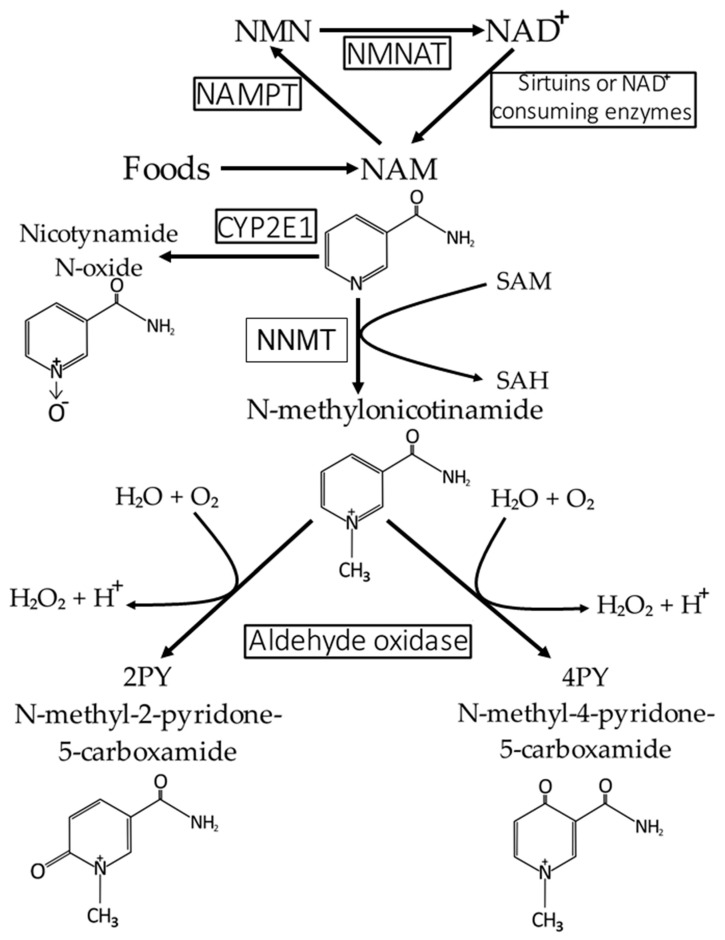
Catabolism of NAD^+^ to 2PY, 4PY, and nicotinamide N-oxide. 2PY—N-methyl-2-pyrdone-5-carboxamide; 4PY—N-methyl-2-4yrdone-5-carboxamide; CYP2E1—cytochrome 2E1 (P450 cytochrome); NAD^+^—nicotinamide adenine dinucleotide; NAM—nicotinamide; NAMPT—nicotinamide phosphoribosyltransferase; NMN—nicotinamide mononucleotide; NMNAT—nicotinamide mononucleotide adenylyltransferase; NNMT—nicotinamide N-methyltranserase; SAH—S-adenosyl-L-homocysteine; SAM—S-adenosyl-L-methionine.

**Figure 5 ijms-26-04463-f005:**
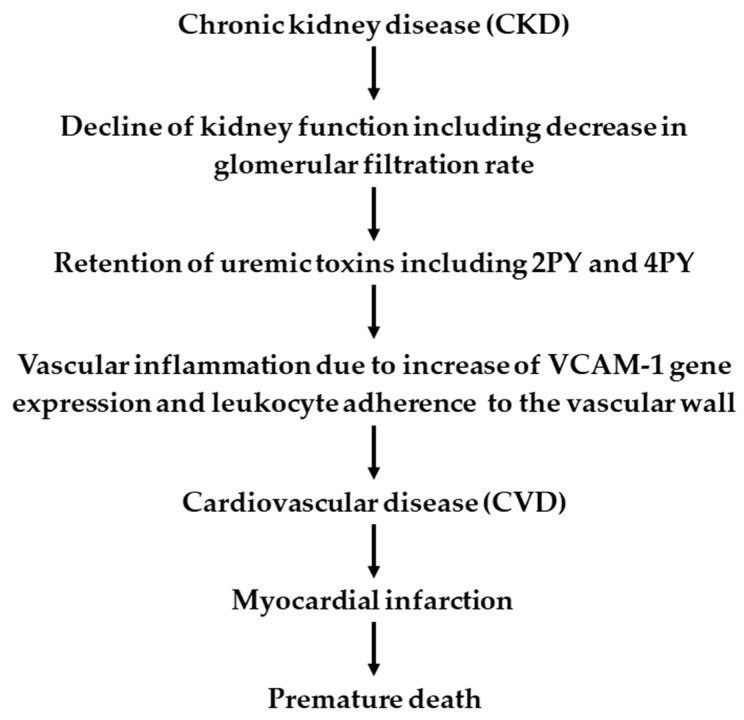
The potential role of elevated serum concentration of 2PY and 4PY on premature death of CKD patients. Based on data presented in [16,18].

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
