# Peer review of "High Concentrations of Circulating 2PY and 4PY—Potential Risk Factor of Cardiovascular Disease in Patients with Chronic Kidney Disease"

_ijms, 2025, doi:10.3390/ijms26094463_

Round 1
Reviewer 1 Report
Comments and Suggestions for Authors
The topic is relevant, timely, and well-grounded, considering the growing interest in NAD⁺ metabolism and its catabolites 2PY and 4PY, particularly in chronic diseases such as chronic kidney disease (CKD). However, some conceptual and structural aspects of the manuscript need to be revised to enhance its scientific rigor and clarity.
-
Excessive focus on biochemical review. The authors dedicate nearly half of the manuscript to an extensive biochemical review of NAD⁺, its biosynthetic pathways, and metabolic functions, with a level of detail more appropriate for a textbook. This extensive background information may distract from the central question of the review, which is the potential relationship between 2PY/4PY, cardiovascular disease (CVD), and CKD. The authors should condense this section and retain only the essential content needed to understand the formation and relevance of these catabolites.
-
Lack of clarity in the central scientific question. Although a hypothesis is mentioned in the introduction, the text does not clearly state the specific knowledge gap the review intends to address. The main research question should be explicitly presented. The authors are encouraged to reformulate the objective as a clear scientific question and, if possible, include a graphical summary of the proposed hypothesis.
-
Need to incorporate more recent literature. The authors should update their review with more recent studies. For example, the following recent publication should be considered:
Ferrell M, Wang Z, Anderson JT, et al. A terminal metabolite of niacin promotes vascular inflammation and contributes to cardiovascular disease risk. Nat Med. 2024 Feb;30(2):424-434.
This study provides important insights into the vascular effects of NAD⁺ catabolites and should be integrated into the discussion.
-
Discussion of “NAD⁺ boosters” lacks balance. The manuscript interestingly explores the topic of NAD⁺ precursors (“boosters”), but it combines efficacy data with possible risks without presenting sufficient quantitative evidence to support clinical concern. A more nuanced and evidence-based approach is needed.
-
Discussion needs restructuring. The discussion should be reorganized to include limitations of the reviewed studies, contradictory findings, and more realistic perspectives on clinical implications.
-
Mechanistic hypothesis figures are missing. The manuscript would benefit from the inclusion of schematic figures illustrating the proposed mechanistic hypothesis. This would help readers better visualize the link between NAD⁺ catabolites, CKD, and CVD risk.
Author Response
Rev . 1:
The topic is relevant, timely, and well-grounded, considering the growing interest in NAD⁺ metabolism and its catabolites 2PY and 4PY, particularly in chronic diseases such as chronic kidney disease (CKD). However, some conceptual and structural aspects of the manuscript need to be revised to enhance its scientific rigor and clarity.
- Excessive focus on biochemical review. The authors dedicate nearly half of the manuscript to an extensive biochemical review of NAD⁺, its biosynthetic pathways, and metabolic functions, with a level of detail more appropriate for a textbook. This extensive background information may distract from the central question of the review, which is the potential relationship between 2PY/4PY, cardiovascular disease (CVD), and CKD. The authors should condense this section and retain only the essential content needed to understand the formation and relevance of these catabolites.
In revised version we have removed most data regarding the role of NAD+ in carbohydrates, lipids and proteins metabolism ( Figs 2 and 3 and Table 1 original version), focusing on NAD+ synthesis and catabolism.
2. Lack of clarity in the central scientific question. Although a hypothesis is mentioned in the introduction, the text does not clearly state the specific knowledge gap the review intends to address. The main research question should be explicitly presented. The authors are encouraged to reformulate the objective as a clear scientific question and, if possible, include a graphical summary of the proposed hypothesis.
We agree with the Rev. 1 that our search question wasn’t explicitly presented in original version. We have reformulated all the basic components of our review.
3. Need to incorporate more recent literature. The authors should update their review with more recent studies. For example, the following recent publication should be considered:
Ferrell M, Wang Z, Anderson JT, et al. A terminal metabolite of niacin promotes vascular inflammation and contributes to cardiovascular disease risk. Nat Med. 2024 Feb;30(2):424-434.
This study provides important insights into the vascular effects of NAD⁺ catabolites and should be integrated into the discussion.
We agree with Rev. 1 that it is an important publication. It was already cited and discussed in original version in position [17], in revised version in position [18].
- Discussion of “NAD⁺ boosters” lacks balance. The manuscript interestingly explores the topic of NAD⁺ precursors (“boosters”), but it combines efficacy data with possible risks without presenting sufficient quantitative evidence to support clinical concern. A more nuanced and evidence-based approach is needed.
We tried to improve this part of our review.
- Discussion needs restructuring. The discussion should be reorganized to include limitations of the reviewed studies, contradictory findings, and more realistic perspectives on clinical implicatins.
Discussion was reconstructed.
- Mechanistic hypothesis figures are missing. The manuscript would benefit from the inclusion of schematic figures illustrating the proposed mechanistic hypothesis. This would help readers better visualize the link between NAD⁺ catabolites, CKD, and CVD risk.
We agree with the Rev. 1 that scheme showing the link between NAD⁺ catabolites, CKD, and CVD risk is necessary. Thus, we have added Figure 5 and graphical abstract
We would like to thank the Reviewer for valuable comments. We hope we have managed to fulfil all the comments and requirements for our manuscript to be published.
Reviewer 2 Report
Comments and Suggestions for Authors
The article “High concentrations of circulating 2PY and 4PY – potential risk factor of cardiovascular disease in patients with chronic kidney disease” is a study regarding 2PY and 4PY as real uremic toxins and possible connection for CVD risk in CKD patients.
The manuscript might be of interest for the readers but contains several flaws that prevent its publication. I was uncertain whether to reject the manuscript or request major revisions; after careful consideration, I have opted for the latter. Authors are required to address the following concerns:
- An extensive English revision of the manuscript is mandatory. Some expressions are unacceptable (e.g. “In summary, one can conclude that …” “However, the study published in Nature Medicine did not address situations…”).
The manuscript is full of mistakes and typos (e.g. Aminocarboxymuconic semialdehyde is written “Aminocarboxymuconic semialhedyde”). It seems it was submitted carelessly without an accurate revision although it was specified in the acknowledgments that it received even an extra revision from a scientist not present in the authors list.
- Line 249 nicotinamide should be before the acronym NAM.
- Due to the topic of the review, authors should reduce all the generic parts regarding the metabolism in mammals, focusing only in recognized human pathways. E.g. “In mammals (specifically in mice), gut microbiota boost host NAD+ metabolism by engaging the deamidated biosynthesis pathway”. Is this information useful for CDK patients?
- I have big concerns about figures whose quality is unacceptable. They do not look like professional, the figures appear more whimsical and informal (e.g. the arrows are irregular, in a cartoon style). Please restyle all figures in a more professional and clearer style. Please confirm that figures are original and generated by authors, and that they are not the result of application of filter or changing parameters (e.g. powerpoint).
- Figure 3 is even cut.
- At page 5 there is a table without caption, and “pathway” is written “pahway”.
- In figure 6 there is a double mistake. In the legend the enzyme Nicotinamide N-methyltranserase is written “NMMT”, in the figure “NMNT”. Both are incorrect, it must be replaced by NNMT which is the universal acronym used for this enzyme.
- The paragraph “NAD synthesis pathways in mammals” lacks of important information. Indeed, NAD content can be modulated also by the extracellular conversion of nicotinamide mononucleotide and nicotinamide riboside by CD73 (PMID: 32389638).
- It has been reported that NNMT activity can greatly impact NAD levels thus influencing Sirtuins activity which utilizes NAD+ as substrate. This is crucial for vascular health and author should acknowledge it (PMID: 39273039).
- The effect of NAD boosters has been recently extensively reviewed thus authors should provide a deeper insight in this topic (PMID: 36829935).
- Due to the high levels of 2PY and 4PY as CVD in CDK (as reported in line 455), it is conceivable to propose the use of NNMT inhibitors for clinical applications. Notably, a number of NNMT inhibitors are already available and were proposed as a promising strategy for several pathological conditions (PMID: 34572571; PMID: 34704059; PMID: 34424711). Please discuss.
- The manuscript contains useless parts for the topic addressed by this review. For instance, lines 557-564 refers to NAM in dermatological pathologies. There is no link to cardiovascular risk or CDK. The same for “It has been also reported that higher niacin intake could have protective effect on the development cognitive decline (Alzheimer disease and other forms of dementia) [62,114].” Please remove all the paragraphs that report information out of focus of the review.
- Please provide a table summarizing studies involving 2PY and 4PY linked to cardiovascular risk or CDK. Please report limitations for each study.
Author Response
Rev 2:
The article “High concentrations of circulating 2PY and 4PY – potential risk factor of cardiovascular disease in patients with chronic kidney disease” is a study regarding 2PY and 4PY as real uremic toxins and possible connection for CVD risk in CKD patients.
The manuscript might be of interest for the readers but contains several flaws that prevent its publication. I was uncertain whether to reject the manuscript or request major revisions; after careful consideration, I have opted for the latter. Authors are required to address the following concerns:
- An extensive English revision of the manuscript is mandatory. Some expressions are unacceptable (e.g. “In summary, one can conclude that …” “However, the study published in Nature Medicine did not address situations…”).
The manuscript is full of mistakes and typos (e.g. Aminocarboxymuconic semialdehyde is written “Aminocarboxymuconic semialhedyde”). It seems it was submitted carelessly without an accurate revision although it was specified in the acknowledgments that it received even an extra revision from a scientist not present in the authors list.
Sorry for all the mistakes. We’ve undertaken one more revision of our English.
- Line 249 nicotinamide should be before the acronym NAM.
We agree with the Rev 2. It was changed in revised version (see Line 206)
- authors should reduce all the generic parts regarding the metabolism in mammals, focusing only in recognized human pathways. E.g. “In mammals (specifically in mice), gut microbiota boost host NAD+ metabolism by engaging the deamidated biosynthesis pathway”. Is this information useful for CDK patients
We agree with the Rew. 2. In revised version the useless information was removed.
4. I have big concerns about figures whose quality is unacceptable. They do not look like professional, the figures appear more whimsical and informal (e.g. the arrows are irregular, in a cartoon style). Please restyle all figures in a more professional and clearer style. Please confirm that figures are original and generated by authors, and that they are not the result of application of filter or changing parameters (e.g. powerpoint).
All the figures were hand painted using Krita.as a tool (on a graphical tablet Huion). Absolutely no image bricks were copied form the web. In revised version I have exchanged arrows in PowerPoint according to your suggestion.
- Figure 3 is even cut.
In revised version Figure 3 was removed according to the suggestion of Rev.1.
6. At page 5 there is a table without caption, and “pathway” is written “pahway”.
In revised version the mentioned Table was removed according to the suggestion of Rev. 1
- In figure 6 there is a double mistake. In the legend the enzyme Nicotinamide N-methyltranserase is written “NMMT”, in the figure “NMNT”. Both are incorrect, it must be replaced by NNMT which is the universal acronym used for this enzyme.
.
In revised version the error was corrected (see Figure 4)
- The paragraph “NAD synthesis pathways in mammals” lacks of important information. Indeed, NAD content can be modulated also by the extracellular conversion of nicotinamide mononucleotide and nicotinamide riboside by CD73 (PMID: 32389638).
In revised version information about extracellular conversion of nicotinamide mononucleotide and nicotinamide riboside by CD73 was included - see line 236-237:
- It has been reported that NNMT activity can greatly impact NAD levels thus influencing Sirtuins activity which utilizes NAD+ as substrate. This is crucial for vascular health and author should acknowledge it (PMID: 39273039).
In revised version this information was added - line 281-283
10. The effect of NAD boosters has been recently extensively reviewed thus authors should provide a deeper insight in this topic (PMID: 36829935).
We have added deeper insight into NAD boosters. We have this paper in our references at position [14].
11. Due to the high levels of 2PY and 4PY as CVD in CDK (as reported in line 455), it is conceivable to propose the use of NNMT inhibitors for clinical applications. Notably, a number of NNMT inhibitors are already available and were proposed as a promising strategy for several pathological conditions (PMID: 34572571; PMID: 34704059; PMID: 34424711). Please discuss.
In revised version NNMT inhibitors were proposed to be used in clinical practice – see lines 425-430.
12. The manuscript contains useless parts for the topic addressed by this review. For instance, lines 557-564 refers to NAM in dermatological pathologies. There is no link to cardiovascular risk or CDK. The same for “It has been also reported that higher niacin intake could have protective effect on the development cognitive decline (Alzheimer disease and other forms of dementia) [62,114].” Please remove all the paragraphs that report information out of focus of the review.
Several papers indicate that clinical use of NAD precursors lead to increase in circulating 2PY and 4PY concentrations – potential link to CVD). Thus, in this review we tried to fit as much information as possible about clinical use of NAD precursors including dermatological pathologies and Alzheimer disease. Moreover, Rev 1 wanted to increase that part of discussion, therefore we are leaving it.
13. Please provide a table summarizing studies involving 2PY and 4PY linked to cardiovascular risk or CDK. Please report limitations for each study.
So far, to our best knowledge, only Ferrell et all [18] showed a relationship between 2PY/4PY and CVD. Circulating high concentration of 2PY/4PY in CKD patients as well as a relationship between CKD and CVD was reported in several papers, That is the base of our hypothesis presented in this review. Thus, we haven’t added the table because in our opinion number of studies regarding the relationship between 2PY/ 4PY and CVD is limited to 1 paper.
We would like to thank the Reviewer for valuable comments. We hope we have managed to fulfil all the comments and requirements for our manuscript to be published.
Round 2
Reviewer 1 Report
Comments and Suggestions for Authors
I have no concerns.
Reviewer 2 Report
Comments and Suggestions for Authors
The manuscript has been revised by authors adressing all the raised concerns and therefore it can be accepted for publication.